# Multi-Spiral Laser Patterning of Azopolymer Thin Films for Generation of Orbital Angular Momentum Light

**DOI:** 10.3390/nano13030612

**Published:** 2023-02-03

**Authors:** Alexey P. Porfirev, Nikolay A. Ivliev, Sergey A. Fomchenkov, Svetlana N. Khonina

**Affiliations:** Image Processing Systems Institute of RAS—Branch of the FSRC “Crystallography and Photonics” RAS, 443001 Samara, Russia

**Keywords:** azopolymer, optical vortex, orbital angular momentum, laser material processing, interference

## Abstract

Recently, the realization of the spiral mass transfer of matter has attracted the attention of many researchers. Nano- and microstructures fabricated with such mass transfer can be used for the generation of light with non-zero orbital angular momentum (OAM) or the sensing of chiral molecules. In the case of metals and semiconductors, the chirality of formed spiral-shaped microstructures depends on the topological charge (TC) of the illuminating optical vortex (OV) beam. The situation is quite different with polarization-sensitive materials such as azopolymers, azobenzene-containing polymers. Azopolymers show polarization-sensitive mass transfer both at the meso and macro levels and have huge potential in diffractive optics and photonics. Previously, only one-spiral patterns formed in thin azopolymer films using circularly polarized OV beams and double-spiral patterns formed using linearly polarized OV beams have been demonstrated. In these cases, the TC of the used OV beams did not affect the number of formed spirals. In this study, we propose to use two-beam (an OV and a Gaussian beam with a spherical wavefront) interference lithography for realization spiral mass transfer with the desired number of formed spirals. The TC of the OV beam allows for controlling the number of formed spirals. We show the microstructures fabricated by the laser processing of thin azopolymer films can be used for the generation of OAM light at the microscale with the desired TC. The experimentally obtained results are in good agreement with the numerically obtained results and demonstrate the potential of the use of such techniques for the laser material processing of polarization-sensitive materials.

## 1. Introduction

The formation of spiral-shaped structures has attracted the attention of researchers in recent years. Such structures formed both at the macro scale and micro scale allow for the generation of optical vortex (OV) beams, structured laser beams with a helical wavefront defined in polar coordinates (*r*, φ) by a phase term exp(*im*φ) where *m* is the topological charge (TC) of the beam [1]. The TC of a light field is closely related to the orbital angular momentum (OAM) of the light field, the value of which is defined as *m*ћ per photon (ћ is the reduced Planck constant) [2]. In addition, spiral-shaped nano- and microstructures can be used for detecting helical dichroism [3].

Various techniques have been used for the fabrication of spiral-shaped nano- and microstructures—direct laser scanning lithography [4,5]; two-photon direct laser writing [6,7,8,9]; the selective electroless metallization of a phospholipid microtubule template [10]; holographic lithography using six linearly polarized side beams and one circular polarized central beam setup [11]; and solvent-cast direct-write (SC-DW) fabrication [12]. Two-photon direct laser writing based on the use of superimposed OV beams and Gaussian beams was used for the fabrication of three-dimensional chiral microstructures with the desired number of lobes whose directions are determined by the sign of the TC [13]. In addition, specially designed multi-ramp helical conical beams have also been used for the formation of chiral multi-lobe microstructures through a two-photon polymerization effect [14]. However, all these techniques require multiple processing steps. By contrast, direct laser machining with structured laser beams is a one-step, high-throughput method. Recently, the spiral-shaped mass transfer of matter and the fabrication of spiral-shaped nano- and microstructures in different materials were realized with OV beams. OVs were used for the formation of spiral-shaped nano- and microneedles in thin films of different materials such as noble metals [15,16], semiconductors [17,18], and polymers [19,20,21,22,23]. It is possible to control the chirality of fabricated nano- and microneedles using OV beams with various TCs as well as multi-spiral light patterns [24].

A recent study showed that in the case of polarization-sensitive polymers such as carbazole-containing azopolymer, laser patterning with OV beams has some limitations. There is no possibility to control the number of the fabricated spiral/lobes of the fabricated chiral structures—nano- and microstructures with one spiral are formed with illuminating circularly polarized radiation and double-spiral structures are formed for linearly polarized radiation. This can be explained by specific distributions of the transverse and longitudinal components of the focused OV beams with different TCs [23]. As it was shown in our previous studies, the longitudinal component of light approximates the profiles of the structures fabricated in azopolymer thin films very well [25]. Currently, systems and elements based on azopolymers are actively used as photoswitches, effecting rapid and reversible control over a variety of chemical, mechanical, electronic, and optical properties [26,27,28,29,30,31]. In this regard, it is important to develop new methods for the low-cost and high-performance laser processing of these materials, including methods for the fabrication of chiral nano- and microstructures with the desired number of fabricated spirals/lobes.

In this study, we adopt the approach used to create multi-spiral interference light distributions for the formation of three-dimensional chiral microstructures using the two-photon direct laser writing method [13] and apply it to realize the multi-spiral laser patterning of azopolymer thin films. The study carried out by Ni et al. [13] demonstrated the use of the interference approach for the shaping of three-dimensional rotating multi-spot laser beams, which was used for the fabrication of three-dimensional chiral microstructures via a combination of the photopolymerization and development of the sample. The fabricated microstructures could be used as meta-atoms for the manufacturing of chiral metamaterials; however, they did not demonstrate application for the generation of OAM beams. By contrast, our proposed approach allows for the generation of two-dimensional spiral-shaped interference laser patterns on the surface of a processed sample that can be used for the realization of spiral-shaped mass transport and the fabrication of three-dimensional multi-spiral microstructures without additional sample deposition procedure. The proposed method allows us to form spiral-shaped microstructures with the desired number of spirals/lobes on the surface of azopolymer thin films. In addition, we show that the fabricated chiral microstructures can be used for the generation of OAM laser beams with TCs which are equal to the number of formed spirals.

## 2. Materials and Methods

### 2.1. Azopolymer Thin Film Preparation

In this study, we used azopolymer thin films fabricated from carbazole-based polymer 9-(2,3-epoxypropyl) carbazole (EPC) and azo dye Disperse Orange 3 (DO3) [32]. The preparation procedure is described in detail in [25] (see Figure 1a). This azopolymer film has a high polarization sensitivity and transmission of about 50% at a wavelength of 532 nm, which was used in this study for direct laser writing. The thickness of the fabricated azopolymer thin films was 1.5 µm.

### 2.2. Spiral-Shaped Intensity Pattern Generation

It is well known that the co-axial interference of OV beams with a TC *m* with a different radial wavefront such as conical [33,34] or spherical [35,36] leads to the formation of a spiral intensity distribution. In particular, intensity for interference of the *m*-th order OV beam with a spherical wavefront can be written as follows:(1)$$I\left(r,\phi \right)={\left|\exp\left(im\phi \right)+\exp\left(i\alpha r^{2}\right)\right|}^{2},$$
where α determines the curvature of the wavefront.

The main characteristic of the interference pattern defined in Equation (1) is related to the function of the phase difference of two wavefronts:(2)$$T\left(r,\phi \right)=\cos\left(m\phi -\alpha r^{2}\right),$$
which determines the spiral nature of the generated interference fringes and results in the generation of *m*-spiral intensity pattern (Figure 2a). This allows one to visually determine the TC of the generated OV beam and has previously been used for the measurement of the mode purity of the generated OV beams. Equation (1) can also be used to determine the intensity extrema on the interferogram. In particular, the boundary between light and dark fringes occurs at the condition: (3)$$m\phi -\alpha r^{2}=\pi \left(q+0.5\right),\;$$
where *q* is an integer.

From Equation (3), one can obtain an explicit form of the spiral dependence of the radius on the angle:(4)$$r(\phi )=\sqrt{\frac{m\phi -\pi \left(q+0.5\right)}{\alpha }}.$$

The generated pattern can be adjusted not only with TC *m* of the used OV beam but also with the wavefront curvature of a Gaussian beam as well as with the ratio of the intensities of the used laser beams. Using the latter parameter, we can completely cancel the interference and form either an annular OV intensity distribution or a Gaussian beam intensity distribution.

For the generation of such interference patterns, usually interferometers with two arms are used—for example, Mach–Zehnder interferometers. However, it is possible to use optical schemas with one arm—for example, [37] presents an approach when such spiral-shaped interference patterns were observed because of the interference of a Gaussian beam with a spherical wavefront transmitted through the central part of a fabricated metasurface and an OV generated with a non-central part of this element. Such an approach can also be realized with one spatial light modulator (SLM)—it is well known that such devices modulate only a part of the incident radiation while some part of the incident radiation does not undergo any modulation and is simply reflected from the display. This part of the radiation propagates along the optical axis. When we use a spherical lens after the SLM, the unmodulated and modulated part of light is focused at the same plane and we can observe the interference between them. However, in this case, we observe the interference of laser beams with plane wavefronts. In order to realize the interference of laser beams with spherical wavefronts we can take the following steps: (1) A spherical phase term should be added to the phase of an element implemented with SLM and generating an OV beam (see the principal schema of the interference with one SLM shown in Figure 2b); (2) the observation plane of the generated interference patterns should be shifted towards the spherical lens at the plane when the generated OV beam is focused. Using such an approach, it is possible to adjust the phase mask parameters (such as a curvature of the added spherical phase term and the intensity ratio of the modulated and unmodulated parts of the incident laser radiation) to observe spiral-shaped patterns with the desired dimensions and contrast.

## 3. Experimental Results

### 3.1. Realization of Multi-Spiral Laser Patterning of Azopolymer Thin Films

For the realization of the proposed approach, we used an optical setup (Figure 3a based on a reflective SLM HOLOEYE PLUTO VIS (1920 × 1080 pixels, pixel size of 8 μm)). In these experiments, we used laser radiation from a continuous-wave solid-state laser source (λ = 532 nm, *P_out_* = 20 mW) for the laser patterning of azopolymer thin films. As was mentioned above, the azopolymer thin films used in the experiments had a transmission of about 50% that allowed the entire volume of the film to be used for the formation of the nano and microreliefs. An output laser linearly polarized Gaussian beam was extended and collimated with a combination of two lenses L1 and L2 with the focal lengths of 25 and 150 mm. The mirrors M1 and M2 were used to direct the laser beam onto the display of the SLM. The 4-f optical system consisting of two lenses L3 and L4 with focal lengths of 500 and 400 mm and a circular diaphragm D was used for the spatial filtering of the laser radiation reflected from the SLM. The mirrors M3, M4, and M5 were used to direct the generated laser beam into the entrance pupil of the first microobjective MO1 (NA = 0.40). This microbjective focused the formed light field on the azopolymer thin film deposited onto a glass substrate S. The phase mask realized with the SLM for the generation of spiral-shaped interference patterns was designed as a superposition of a spiral phase plate with the desired TC and a spherical phase term. Then, the surface of the used thin film was located at the plane before the focal plane of the microobjective MO1. As was mentioned above, in this case, the surface of the film is located in the plane when spiral-shaped interference patterns are observed. To move the glass substrate, we mounted it on a three-axis XYZ translation stage. The adjustment of the curvature of the adding spherical phase term allowed us to fine tune the interference and form spiral-shaped patterns with the desired dimensions and contrast. To tune a ratio of the intensities of the modulated and non-modulated parts of the laser radiation reflected from the SLM, standard modulator control software was used. A system consisting of a light bulb IB, a spherical lens L6 (focal length of 50 mm), a mirror M6, and a microobjective MO2 (NA = 0.1) was used to illuminate the surface of the glass substrate. To observe the surface of the glass substrate with the illuminated azopolymer thin film during laser patterning, we used a system consisting of a beam splitter BS, lens L5 with a focal length of 150 mm, and a video camera CAM (TOUPCAM UHCCD00800KPA; 1600 × 1200 pixels with a pixel size of 3.34 μm). A neutral density filter F was used to decrease the intensity of the observed light field. For the transformation of the linearly polarized laser radiation into circularly polarized radiation, the quarter-wave plate QWP was used. Figure 3b shows the experimentally generated spiral-shaped interference patterns used in the experiments for the fabrication of one-, two-, three-, and five-spiral microrelief. These intensity distributions were of good quality and were in good agreement with the numerically obtained distributions shown in Figure 2a. For the fabrication of the multi-spiral microrelief, we illuminated the azopolymer thin films with the formed interference patterns for 5 s. The estimated exposure laser power on the polymer thin film is about 1 mW. At such laser power conditions, the affected area of the azopolymer thin films is determined by the transverse size of the beam at the focus [38]. Therefore, no heat-affected zone formation was observed.

The images of the profiles of the microstructures fabricated with circularly polarized radiation and measured with a scanning probe microscope NT-MDT SOLVER Pro-M are shown in Figure 3c. In this case, a semi-contact operating mode was used, which is characterized by the minimal mechanical effect of a silicon probe on the sample surface (line scanning frequency was 0.5 Hz with a study area of 30 × 30 μm). The profiles of the fabricated microstructures coincided well with the generated multi-spiral intensity patterns. The number of the formed spirals on the surface of the used azopolymer thin films and the number of the light spirals of the generated interference patterns were equal. The heights of the microstructures presented in Figure 3c ranged from 150 nm in the case of the five-spiral microrelief up to 850 nm in the case of the one-spiral microrelief. These values are noticeably smaller than the thickness of the used azopolymer thin film, which was 1500 nm. Because of this, no influence of the glass substrate on the properties of the patterned azopolymer films was observed. In addition, we did not detect any modification of the film interface-substrate or on the glass surface. This was most likely caused by the relatively low power of the laser radiation used. It should be noted that an increase in laser power resulted in an increase in the height of the “head” of the shaped spiral. Earlier, such dependence of the height of the fabricated microstructures on the surface of carbazole containing azopolymer thin films was demonstrated for laser radiation with various wavelengths [19,39]. In our case, since the transfer of the temporarily melted azopolymer under the action of laser radiation occurs in the direction of the intensity gradient, this behavior can be explained by the increase in optical gradient forces with increasing laser power [40].

The formation of structures in azopolymer thin films strongly depends on the polarization state of the illuminating laser beam. In our experiments, multi-spiral microstructures were obtained with both linearly and circularly polarized laser beams. However, in the case of circularly polarized radiation, more symmetrical distributions were formed. To explain the similarity of the microstructure profiles formed in these two cases, some different characteristics of the generated interference patterns were investigated using the Debye approximation and Richards–Wolf formulas [41] in the following form:(5)$$\begin{array}{l} E(\rho ,\psi ,z)=\left(\begin{array}{c} E_{x}(\rho ,\psi ,z)\\ E_{y}(\rho ,\psi ,z)\\ E_{z}(\rho ,\psi ,z) \end{array}\right)=-\frac{if}{\lambda }\times \\ \times \int_{0}^{\Theta }\int_{0}^{2\pi }\left(\begin{array}{ll} \left[1+\cos^{2}v(\cos\theta -1)\right] & \sin v\cos v(\cos\theta -1)\\ \sin v\cos v(\cos\theta -1) & \left[1+\sin^{2}v(\cos\theta -1)\right]\\ -\sin\theta \cos v & -\sin\theta \sin v \end{array}\right)\left(\begin{array}{c} c_{x}(v)\\ c_{y}(v) \end{array}\right)\times \\ \times B(\theta ,v)T(\theta )\exp\left[ik(\rho \sin\theta \cos(v-\psi )+z\cos\theta )\right]\sin\theta \;d\theta \;dv\;, \end{array}$$
where (ρ, ψ, *z*) are the cylindrical coordinates in the focal region, (θ, *ν*) are the spherical angular coordinates of the focusing system’s output pupil, Θ is the maximum value of the azimuthal angle related to the system’s numerical aperture, *B*(θ, *ν*) is the transmission function, *T*(θ) = (cosθ)^1/2^ is the pupil’s apodization function of aplanatic systems, *f* is the focal length, and ***C***(*ν*) = (*c_x_*(*ν*), *c_y_*(*ν*))*^T^* is the polarization vector.

In our investigation, we studied the intensity of the longitudinal component of the generated light fields |*E_z_*|^2^ and the inversion of the intensity of the longitudinal component defined as ∇^2^|*E_z_*|^2^ as well as the relief height distribution calculated from the model proposed by Ambrosio et al. in 2012 for the description of light-induced spiral mass transport in azopolymer films under OV beam illumination [19]. With this model, the surface ‘height’ variations across the polymer film were calculated as follows:(6)$$h_{m}(x,y)\propto \left(c_{1}+c_{2}\right)M_{0}+c_{1}M_{1}+c_{2}M_{2}+c_{3}M_{3}+c_{B}M_{B},$$
where $M_{0}=\frac{\partial^{2}}{\partial x^{2}}{\left|E_{x}\right|}^{2}+\frac{\partial^{2}}{\partial y^{2}}{\left|E_{y}\right|}^{2}$, $M_{1}=\frac{\partial^{2}}{\partial x^{2}}{\left|E_{y}\right|}^{2}+\frac{\partial^{2}}{\partial y^{2}}{\left|E_{x}\right|}^{2}$, $M_{2}=\frac{\partial }{\partial x}\frac{\partial }{\partial y}\left(E_{y}^{*}E_{x}+E_{x}^{*}E_{y}\right)$, $M_{3}=\nabla^{2}{\left|E_{z}\right|}^{2}=\frac{\partial^{2}}{\partial x^{2}}{\left|E_{z}\right|}^{2}+\frac{\partial^{2}}{\partial y^{2}}{\left|E_{z}\right|}^{2}$, and MB=∂∂xRe(Ez*Ex)+∂∂yRe(Ez*Ey).

Previously [42], it was shown that the inversion of the intensity of the longitudinal component and the sum of two terms *M*_0_ + *M*_2_ ∝ Re[∇F_p_], which is close to the value of the divergence of the polarization force [19], have a significant effect on the structure of the relief experimentally formed in thin azopolymer films. Figure 4 shows the results of the calculation of these characteristics. 

As can be seen from the modeling results shown in Figure 4, the state of polarization definitely affects the intensity pattern of the longitudinal component |*E_z_*|^2^, although the distribution of the total intensity |**E**|^2^ in this case remains unchanged for the same values of the TC due to the small value of the numerical aperture of the focusing lens [43]. Note that the considered characteristics, such as ∇^2^|*E_z_*|^2^ which is regarded in [19] and Re[∇F_p_] which is used in [25,42] for predicting the form of the relief height distribution, also correlate significantly with the |*E_z_*|^2^ structure. It can be seen that circular polarization provides a clearer and more symmetrical distribution. It should also be noted that in the considered case of uniform polarization, the formed relief and the characteristics predicting its shape are close to the distribution of the total intensity.

### 3.2. Generation of OAM Light Using the Fabricated Spiral-Shaped Microstructures

The profiles of the fabricated spiral-shaped microstructures are similar to the profiles of spiral axicons and spiral Fresnel zone plates, diffractive optical elements which are widely used for the generation of OAM laser beams [1]. To verify that the fabricated elements generate OAM laser beams, an optical setup based on the Mach–Zehnder interferometer was used (Figure 5). The output linearly polarized Gaussian laser beam from a He-Ne laser source (λ = 633 nm, P_out_ = 1 mW) was collimated and extended with a PH pinhole (aperture size 40 μm) and lens L1 (focal length 250 mm). A circular diaphragm D1 was used to adjust the laser beam diameter. Then, the collimated laser beam was split into two equivalent laser beams using a BS1 beam splitter. In the object arm of the interferometer, the laser beam was focused by a microobjective MO1 (3.7×, NA = 0.1) onto the surface of the glass substrate with the fabricated multi-spiral microstructures in an azopolymer thin film (P). Using a three-axis XYZ translation stage, we moved the glass substrate and investigated different fabricated microstructures. A microobjective MO2 (4×, NA = 0.1), lenses L2 and L3 with focal lengths of 150 and 100 mm, and a circular diaphragm D2 were used for the imaging and spatial filtering of the generated light fields. A neutral density filter F was used to adjust the intensities of the object and the reference beams. In the reference arm of the interferometer, lens L4 with a focal length of 150 mm was used for the transformation of the laser beam with a plane wavefront into a beam with a spherical wavefront. A second beam splitter BS2 was used to superimpose the object and reference laser beams. A video camera CAM (TOUPCAM UHCCD00800KPA) was used to capture the images of the generated intensity distributions and interference fringes. The distributions of light fields generated experimentally using three different multi-spiral microstructures fabricated in azopolymer thin films are shown in Figure 5b. 

The formed spiral-shaped interference fringes show the presence of a helical wavefront in the generated annular light fields. In this case, the number of formed light spirals indicates the value of the TC of the generated OV beam. Furthermore, the TC of the generated OV beams can be determined from the central part of the generated annular intensity distributions. They clearly show the patterns corresponding to the astigmatic transformation of an OV beam. It is well known that during the astigmatic transformation of an optical vortex with a TC of *m*, the annular intensity distribution transforms into a set of *m* intensity nulls on a line. The quality of the generated interference patterns decreases as the number of formed spirals increases because the height of the formed patterns decreases and is not enough to completely modulate the incident laser beam. Thus, the observed light field is a superposition of the unmodulated part of the radiation and a formed OV beam. It is therefore very important to accurately tune the parameters of the laser beam and exposure time during the laser patterning, which will be the subject of our future research.

## 4. Discussion

Our results demonstrate the possibility of the use of an interferometric approach for manufacturing such rather complex structures as multi-spiral microstructures in azopolymer thin films which are very sensitive to the polarization state of illuminating radiation. This polarization sensitivity does not allow the realization of spiral mass transfer with more than two spirals even when OV beams with high TCs are used. This was demonstrated in our previous study [23]. However, the proposed interferometric approach in the current study made it possible to overcome this limitation and control the number of formed spirals. The fabricated multi-spiral elements are compact and allow the generation of OVs at the microscale. Such an easy-to-implement fabrication technique does not require multi-step processing and ensures the formation of a set of such elements generating OVs at different locations.

Such an approach was realized only with a single spatial light modulator and did not require the use of interferometric schemas. The possibility of controlling the parameters of the generated phase mask directly using standard software allows one to control the parameters of the generated interference fringes. Thus, the profiles of the fabricated microstructures can be adjusted with no modification of the optical setup. The next step in our future investigation will be the study of the possibilities of using phase masks of other complex diffractive optical elements for implementing the direct laser patterning of azo-polymer thin films. 

The calculated characteristics of the formed interference patterns in the cases of linear and circular polarization show certain dependence of the profiles of the fabricated structures on the polarization state. In particular, circular polarization in this regard provides a clearer and more symmetrical distribution. However, in this case (for homogeneous polarization) the structure does not change very noticeably, and it is very close to the total intensity distribution. This situation is similar to the interferometric laser recording of polarization holographic gratings in azopolymer-based materials when only the interference contrast changes for different combinations of the polarization states of laser beams in different interferometer arms. Because of this, it is interesting to investigate other polarization states of the generated interferometric fringes used for laser pattering—for example, higher-order cylindrical vector beams (CVBs). It is known that the interaction of polarization singularities is associated with CVBs and that phase singularities are associated with OV beams [44]. Thus, new results can be obtained in this case.

## 5. Conclusions

An approach for the fabrication of multi-spiral microstructures in azopolymer thin films with direct laser patterning was realized. The approach was based on the use of structured laser beams in the form of spiral-shaped interference fringes generated through the interference of an OV beam with a Gaussian beam with a spherical wavefront. In contrast with demonstrated earlier spiral-shaped mass transfer with OV beams, this technique allows for the fabrication of spiral-shaped microstructures with a desired number of spirals. In addition, there is no significant difference in the profiles of the fabricated multi-spiral microstructures when we use linearly or circularly polarized radiation. The profiles of the fabricated microstructures measured with a scanning probe microscope were very well approximated with the inversion of the intensity of the longitudinal component of an illuminating laser beam, as was noted in our previous studies [23,25]. Here, we further used this approximation to predict the profiles of the formed microstructures and demonstrate the possibility of realization control multi-spiral-shaped mass transfer. A numerical analysis showed that the state of homogeneous polarization does not have a very noticeable effect on the intensity pattern of the longitudinal component |*E_z_*|^2^ and characteristics, such as ∇^2^|*E_z_*|^2^ and Re[∇F_p_], which are used for predicting the form of the relief height distribution, although circular polarization in this case provides a clearer distribution close to the total intensity |**E**|^2^ than does linear polarization. The fabricated multi-spiral microstructures were used for the generation of OV beams with the desired TCs corresponding to the number of formed spirals. The proposed method provides the possibility of the use of structured interference patterns generated with a single spatial light modulator for the fabrication of microelements generating OAM light at the microscale for laser manipulation, integral optics, and biophotonic applications. Thus, our results represent a pathway towards the easy-to-implement inexpensive fabrication of spiral-shaped microstructures for advanced photonic applications. We believe that the fabricated structures can also be used for sensing chiral molecules and controlling helical dichroism. The control of light–matter interactions in these cases may present new opportunities with regard to chiroptical spectroscopy, light-driven molecular machines, optical switching, and the in situ ultrafast probing of chiral systems and magnetic materials [45].

## Figures and Tables

**Figure 1 nanomaterials-13-00612-f001:**
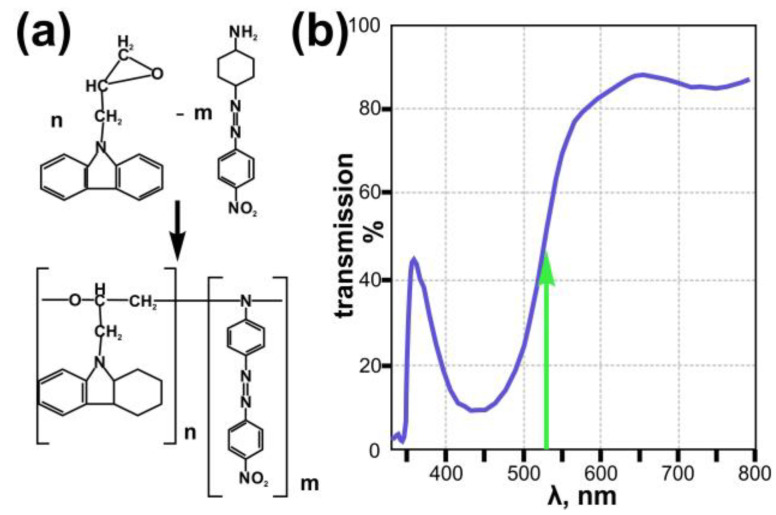
Scheme of synthesis and chemical structure of the used azopolymer EPC:DO3 (**a**) and the transmission spectra of the EPC:DO3 azopolymer film used in this study (**b**). Green arrow indicates wavelength of recording laser (532 nm) used in this study.

**Figure 2 nanomaterials-13-00612-f002:**
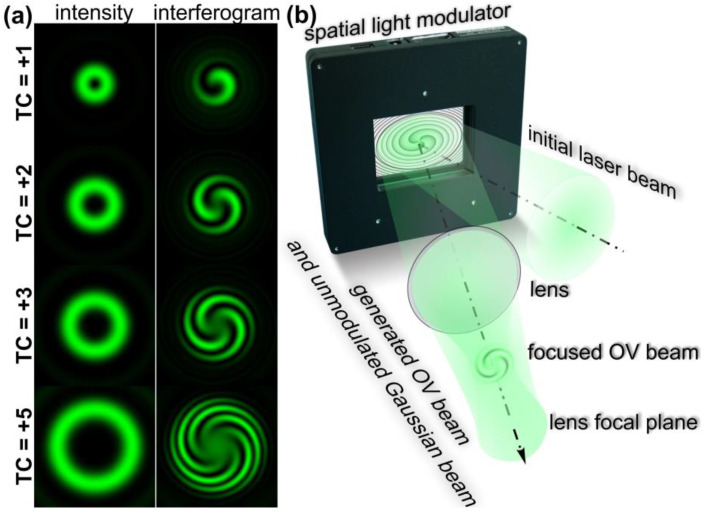
Generation of multi-spiral light interference fringes. (**a**) Intensity distributions of OV beams with different TCs and interference fringes generated as results of the interference of these OV beams with a Gaussian beam with a spherical wavefront. (**b**) Principle of generation of the multi-spiral interference fringes using a single spatial light modulator—multi-spiral light patterns are generated because of the interference of the non-modulated part of the initial Gaussian beam incident on the display of the modulator and a focused OV beam formed with a phase mask realized with the modulator. The generation of multi-spiral interference fringes is observed at the plane before the focal plane of the used lens, where the Gaussian beam with a spherical wavefront is formed.

**Figure 3 nanomaterials-13-00612-f003:**
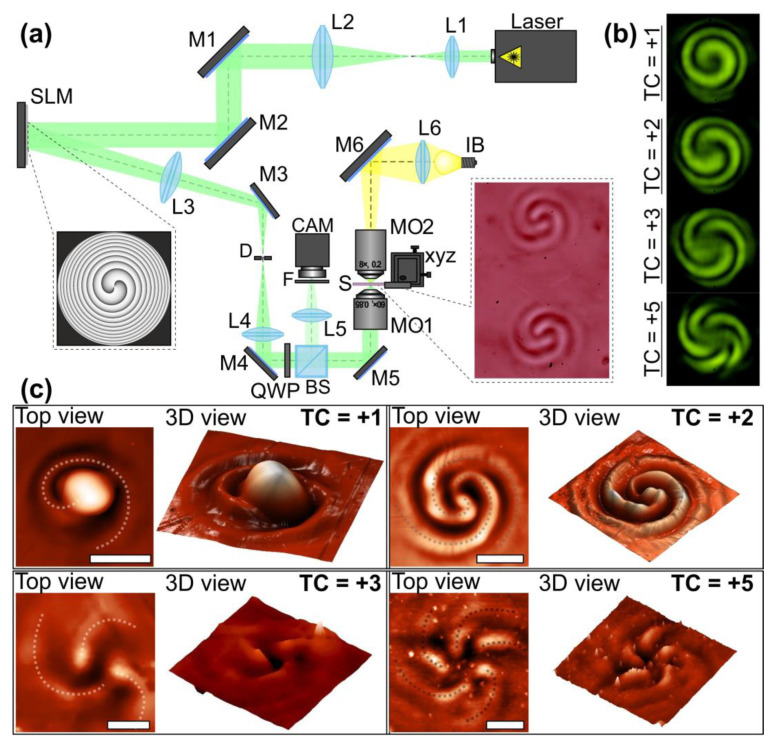
Multi-spiral laser patterning of azopolymer thin films. (**a**) Experimental setup for laser patterning of azopolymer thin films. Laser is a solid-state laser (λ = 532 nm, *P*_out_ = 20 mW); L1, L2, L3, L4, L5, and L6 are spherical lenses (*f*_1_ = 25 mm, *f*_2_ = 150 mm, *f*_3_ = 500 mm, *f*_4_ = 400 mm, *f*_5_ = 150 mm, and *f*_6_ = 50 mm); M1, M2, M3, M4, M5, and M6 are mirrors; SLM is a reflective spatial light modulator (HOLOEYE PLUTO VIS, 1920 × 1080 pixels, pixel size of 8 μm); BS is a beam splitter; MO1 and MO2 are microobjectives (NA = 0.4 and 0.1); S is a glass substrate with an azopolymer thin film that was mounted on the 3-axis XYZ translation stage; IB is a light bulb; F is a neutral density filter; CAM is a video camera (TOUPCAM UHCCD00800KPA; 1600 × 1200 pixels with a pixel size of 3.34 μm); and QWP is a quarter wave plate. The inset on the left side shows a phase of the element generating an OV beam with a TC of 2. The inset on the right side shows an optical microscopy image of the surface of an azopolymer thin film patterned with this OV beam. (**b**) Experimentally obtained multi-spiral interference patterns formed with the described experimental setup in the case of different OV beams. (**c**) Images of the profiles of the microstructures fabricated with circularly polarized radiation and measured with a scanning probe microscope. Four different microstructures fabricated using OV beams with TCs +1, +2, +3, and +5 are shown. The dotted curves indicate the formed spirals. The scale bar is 10 µm.

**Figure 4 nanomaterials-13-00612-f004:**
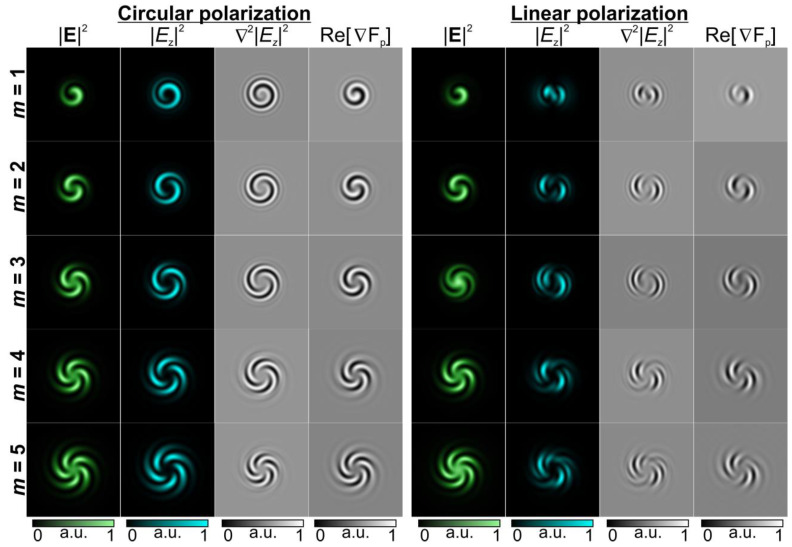
Numerically calculated characteristics of the light fields formed as a result of the interference of a Gaussian laser beam with a spherical wavefront and a circularly/linearly polarized OV beam with a non-zero TC.

**Figure 5 nanomaterials-13-00612-f005:**
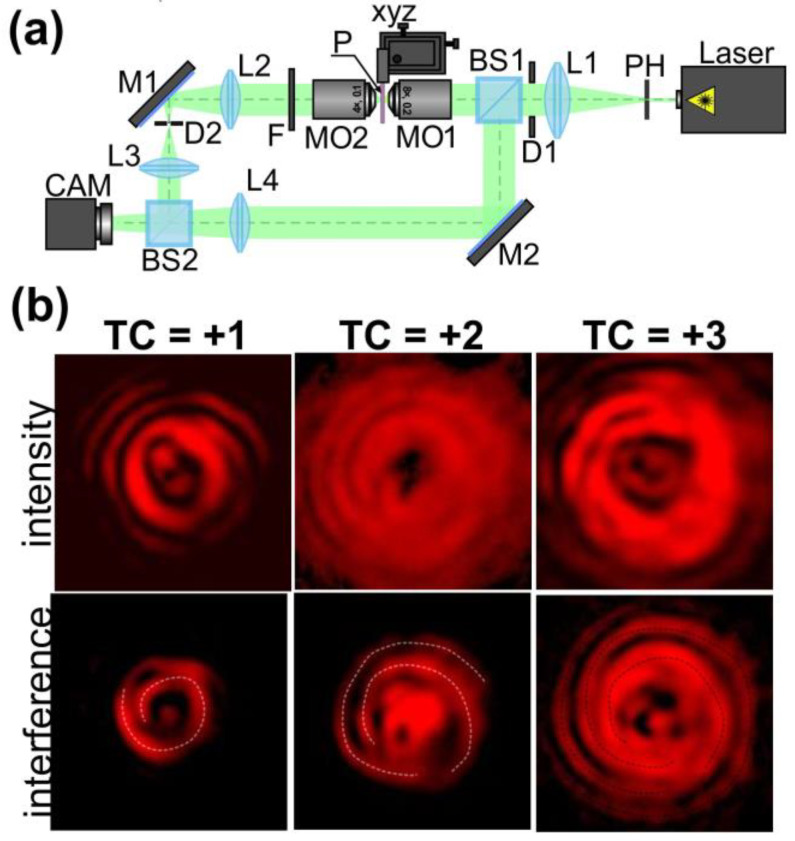
Generation of OAM laser beams with the multi-spiral microstructures fabricated using direct multi-spiral laser patterning in azopolymer thin films. (**a**) The experimental setup for the investigation of the light field formed because of the diffraction of a linearly polarized Gaussian beam on the fabricated microstructures. Laser is a He-Ne (λ = 633 nm, *P*_out_ = 1 mW); PH is a pinhole (aperture size 40 μm); L1, L2, L3, and L4 are spherical lenses (*f*_1_ = 250 mm, *f*_2_ = 150 mm, *f*_3_ = 100 mm, *f*_4_ = 150 mm); M1 and M2 are mirrors; D1 and D2 are circular diaphragms; BS1 and BS2 are beam splitters; MO1 and MO2 are microobjectives (3.7×, NA = 0.1 and 4×, NA = 0.1); P is a glass substrate with fabricated multi-spiral microstructures in an azopolymer thin film; F is a neural density filter; and CAM is a video camera (TOUPCAM UHCCD00800KPA; 1600 × 1200 pixels with a pixel size of 3.34 μm). (**b**) Intensity distributions and interference fringes generated at a distance of 30 µm from the surface of the used azopolymer thin film. Three different multi-spiral microstructures fabricated using OVs with TCs +1, +2, and +3 were investigated. The dotted curves indicate formed spiral interference fringes. The image dimensions are 50 × 50 µm^2^.

## Data Availability

Not applicable.

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
