# Peer review of "Multi-Spiral Laser Patterning of Azopolymer Thin Films for Generation of Orbital Angular Momentum Light"

_nanomaterials, 2023, doi:10.3390/nano13030612_

Round 1

Reviewer 1 Report

The authors Alexey P. Porfirev, Nikolay A. Ivliev, Sergey A. Fomchenkov, Svetlana N. Khonina, of the manuscript titled "Multi-spiral laser patterning of azopolymer thin films for generation of orbital angular momentum light" represented interesting results in the field of patterning the optically semi-transparent azopolymer films at the wavelength of 532 nm by using two-beams (an OV and a Gaussian beam with a spherical wavefront) interference lithography for realization spiral-mass transfer with the desired number of the formed spirals via applying laser radiation from a solid-state laser source (λ = 532 nm, Pout = 20 mW).

The manuscript is well written, but can be published after major revision. There are some details, which are very needed to be specified:

1. So:

             -The authors clearly should specify the novelty of their work, to distinguish very well what their contribution is in comparison with the work they cited, namely Reference 13 -

Ni, J.; Wang, C.; Zhang, C.; Hu, Y.; Yang, L.; Lao, Z.; Xu, B.; Li, J.; Wu, D.; Chu, J. Three-Dimensional Chiral Microstruc-388 tures Fabricated by Structured Optical Vortices in Isotropic Material. Light Sci. Appl. 2017, 6, e17011,

(lines 72-79)

            Since the authors Jincheng Ni, Chaowei Wang, Chenchu Zhang, Yanlei Hu, Liang Yang, Zhaoxin Lao, Bing Xu, Jiawen Li, Dong Wu and Jiaru Chu of the Reference 13 – “Three-Dimensional Chiral Microstruc-388 tures Fabricated by Structured Optical Vortices in Isotropic Material.” in Light Sci. Appl. 2017, 6, e17011, also claimed they “demonstrate 3D chiral microstructures with large area and high precision fabricated in isotropic polymer by developing the coaxial interference of OAM beams and plane waves for the first time….”

and

“…First, we design interfering vortex holograms (IVHs) that can generate interfering vortex beams (IVBs) of optical vortices and plane waves using a liquid-crystal spatial light modulator (SLM)…..”

            - Also, they should distinguish and specify what exactly makes new and/or different this manuscript in comparison with their previous works Reference 23 and mainly Refernce 25, both published in 2022.

            - It means, in the corresponding paragraphs, for instance in “Introduction” and “Conclusions” they should explain clearly their contributions in the 3D microstructures formation by using optical vortex beam.

2. The authors should give more information about the laser beam parameters such as pulse duration (e.g. nano- or femtosecond laser), repetition rate applied in the experiments.

3. Can the authors explain if the repetition rate and the pulse duration of the laser beam and the laser power influence on the 3D microstructures formation by using optical vortex beam and what about this dependence?

4.  What was the thickness of the azopolymer films used in the experiments?

5. Did the authors notice heat affected zone (HAZ) formation beside the local area of the laser impact on the film?

6. Did the authors notice some influence of the glass substrate on the properties of the patterned azopolymer films during the laser patterning by the OV beam and later, for instance some modification and/or activation of the opposite surface of the film, namely of the film interface-substrate or on the glass surface? This could happen when an optically transparent film deposited on glass or other type of substrate is irradiated by laser beam. Even more sometimes this is the desired effect, but sometimes it is strongly unwanted effect.

Author Response

We are thankful to reviewers for their useful comments and suggestions, which allow us to improve the quality of the manuscript making it more clear for readers. We believe the corrections made address the Reviewer’s concerns making the manuscript suitable for publication in the journal.

All changes in the manuscript are highlighted by blue color.

Reviewer 1

The authors Alexey P. Porfirev, Nikolay A. Ivliev, Sergey A. Fomchenkov, Svetlana N. Khonina, of the manuscript titled "Multi-spiral laser patterning of azopolymer thin films for generation of orbital angular momentum light" represented interesting results in the field of patterning the optically semi-transparent azopolymer films at the wavelength of 532 nm by using two-beams (an OV and a Gaussian beam with a spherical wavefront) interference lithography for realization spiral-mass transfer with the desired number of the formed spirals via applying laser radiation from a solid-state laser source (λ = 532 nm, Pout = 20 mW).

The manuscript is well written, but can be published after major revision. There are some details, which are very needed to be specified:

  1. So:

             -The authors clearly should specify the novelty of their work, to distinguish very well what their contribution is in comparison with the work they cited, namely Reference 13 -

Ni, J.; Wang, C.; Zhang, C.; Hu, Y.; Yang, L.; Lao, Z.; Xu, B.; Li, J.; Wu, D.; Chu, J. Three-Dimensional Chiral Microstruc-388 tures Fabricated by Structured Optical Vortices in Isotropic Material. Light Sci. Appl. 2017, 6, e17011,

(lines 72-79)

            Since the authors Jincheng Ni, Chaowei Wang, Chenchu Zhang, Yanlei Hu, Liang Yang, Zhaoxin Lao, Bing Xu, Jiawen Li, Dong Wu and Jiaru Chu of the Reference 13 – “Three-Dimensional Chiral Microstruc-388 tures Fabricated by Structured Optical Vortices in Isotropic Material.” in Light Sci. Appl. 2017, 6, e17011, also claimed they “demonstrate 3D chiral microstructures with large area and high precision fabricated in isotropic polymer by developing the coaxial interference of OAM beams and plane waves for the first time….”

and

“…First, we design interfering vortex holograms (IVHs) that can generate interfering vortex beams (IVBs) of optical vortices and plane waves using a liquid-crystal spatial light modulator (SLM)…..”

            - Also, they should distinguish and specify what exactly makes new and/or different this manuscript in comparison with their previous works Reference 23 and mainly Refernce 25, both published in 2022.

            - It means, in the corresponding paragraphs, for instance in “Introduction” and “Conclusions” they should explain clearly their contributions in the 3D microstructures formation by using optical vortex beam.

Reply:

Thank you for the comment. We added the following text in the Introduction to explain the difference between our results and the results presented in Ref. 13:

The study carried out by Ni et al shows the use of the interference approach for shaping of three-dimensional rotating multi-spot laser beams, which was used for the fabrication of three-dimensional chiral microstructures via a combination of the photopolymerization and development of the sample. The fabricated microstructures can be used as meta-atoms for manufacturing of chiral metamaterials, however, they have not demonstrated application for generating of OAM beams. In contrast, our proposed approach allows one to generate two-dimensional spiral-shaped interference laser patterns on the surface of a processed sample that can be used for realization spiral-shaped mass transport and fabrication of three-dimensional multi-spiral microstructures without additional sample deposition procedure.

We changed the text in the Discussion to clearly difference from Ref. 23. Results in [23] show that it is impossible to realize spiral-mass transfer with more than two spirals when OV beams with high TCs are used. However, the interferometric approach proposed in the current study made it possible to overcome this limitation and control the number of the formed spirals.

Reference 25 only shows that the longitudinal component of light very well approximates the profiles of the structures fabricated in azopolymer thin films. This reference does not related with the realization of spiral-mass transfer. We changed text in the Conclusions to clarify this.

  1. The authors should give more information about the laser beam parameters such as pulse duration (e.g. nano- or femtosecond laser), repetition rate applied in the experiments.

Reply:

In our experiments, we used a continuous-wave laser source. In addition, we added the following information:

The estimated exposure laser power on the polymer thin film is about 1 mW.

  1. Can the authors explain if the repetition rate and the pulse duration of the laser beam and the laser power influence on the 3D microstructures formation by using optical vortex beam and what about this dependence?

Reply:

As we mentioned above, we used a continuous wave laser radiation.

We added the following information:

It should be noted that an increase in laser power resulted in an increase in the height of the “head” of the shaped spiral. Earlier, such dependence of the height of the fabricated microstructures on the surface of carbazole containing azopolymer thin films was demonstrated for laser radiation with various wavelengths [19, 39]. In our case, since the transfer of the temporarily melted azopolymer under the action of laser radiation occurs in the direction of the intensity gradient, this behavior can be explained by the increase in optical gradient forces with increasing laser power [40].

[39] Podlipnov, V.V., Ivliev, N.A., Khonina, S.N., Nesterenko, D.V., Meshalkin, A.Y. and Achimova, E.A., 2019, November. Formation of microstructures on the surface of a carbaseole-containing azopolymer by the action of laser beams. In Journal of Physics: Conference Series (Vol. 1368, No. 2, p. 022069). IOP Publishing.

[40] Kumar, J., Li, L., Jiang, X.L., Kim, D.Y., Lee, T.S. and Tripathy, S., 1998. Gradient force: The mechanism for surface relief grating formation in azobenzene functionalized polymers. Applied Physics Letters, 72(17), pp.2096-2098.

  1. What was the thickness of the azopolymer films used in the experiments?

Reply:

The thickness of the fabricated azopolymer thin films was 1.5 µm. We added this information

  1. Did the authors notice heat affected zone (HAZ) formation beside the local area of the laser impact on the film?

Reply:

At the small laser power used in our experiments, the affected area of the azopolymer thin films is determined by the transverse size of the beam at the focus. Because of this, we did not notice HAZ formation. We added the following text:

At such laser power conditions, the affected area of the azopolymer thin films is determined by the transverse size of the beam at the focus [38]. Therefore, there is no heat-affected zone formation was observed.

[38] Miniewicz, A., Sobolewska, A., Piotrowski, W., Karpinski, P., Bartkiewicz, S. and Schab-Balcerzak, E., 2020. Thermocapillary Marangoni flows in azopolymers. Materials, 13(11), p.2464.

  1. Did the authors notice some influence of the glass substrate on the properties of the patterned azopolymer films during the laser patterning by the OV beam and later, for instance some modification and/or activation of the opposite surface of the film, namely of the film interface-substrate or on the glass surface? This could happen when an optically transparent film deposited on glass or other type of substrate is irradiated by laser beam. Even more sometimes this is the desired effect, but sometimes it is strongly unwanted effect.

Reply:

The thickness of the used azopolymer thin films is 1500 nm, while the maximum height of the fabricated multi-spiral relief is 850 nm. Thus, we did not notice the influence of the glass substrate on the properties of the patterned azopolymer films. We added the following information:

These values are noticeably smaller that the thickness of the used azopolymer thin film. Because of this, no influence of the glass substrate on the properties of the patterned azopolymer films was observed.

Reviewer 2 Report

1. there is one typo in the 226th line, "if" should be "in"

2. What is the height of the fabricated spiral structure? Like in Fig. 2(c).

3. The generated vortex beams and interferences in Fig.4 are not of good quality.  The authors should discuss the reason for that and possible improvements for commercialization.

4. What is the efficiency of this method for generating OV?

After addressing the above concerns, I would recommend the publication of this manuscript.

Author Response

We are thankful to reviewers for their useful comments and suggestions, which allow us to improve the quality of the manuscript making it more clear for readers. We believe the corrections made address the Reviewer’s concerns making the manuscript suitable for publication in the journal.

All changes in the manuscript are highlighted by blue color.

Reviewer 2

  1. there is one typo in the 226th line, "if" should be "in"

Reply:

Thank you. We corrected it.

  1. What is the height of the fabricated spiral structure? Like in Fig. 2(c).

Reply:

We added the following information:

The height of the microstructures presented in Fig. 2 (c) is in the range from 150 nm in the case of 5-spiral microrelief up to 850 nm in the case of 1-spiral microrelief.

  1. The generated vortex beams and interferences in Fig.4 are not of good quality. The authors should discuss the reason for that and possible improvements for commercialization.

Reply:

The quality of the generated interference patterns decreases as the number of formed spirals increases. This can be explained by the fact that, with multispiral reliefs, the height of the formed patterns is not enough to completely modulate the incident laser beam. Thus, the observed light field is a superposition of the unmodulated part of the radiation and a formed OV beam. Therefore, it is very important to accurately tune the parameters of the laser beam and the exposure time during the laser patterning which is our future research. We added this text.

  1. What is the efficiency of this method for generating OV?

Reply:

The fabricated multi-spiral elements are compact and allow one to generate OVs at microscale. Such easy-to-implement fabrication technique does not require multi-step processing and ensures the formation of a set of such elements generating OVs at different locations. However, the generation of OVs is only one of the applications of the proposed two-beam interference lithography. As we mentioned in conclusions, the fabricated structures potentially can be used for sensing of chiral molecules and control of helical dichroism.

We added the following information:

The fabricated multi-spiral elements are compact and allow one to generate OVs at microscale. Such easy-to-implement fabrication technique does not require multi-step processing and ensures the formation of a set of such elements generating OVs at different locations.

Round 2

Reviewer 1 Report

The authors Alexey P. Porfirev, Nikolay A. Ivliev, Sergey A. Fomchenkov, Svetlana N. Khonina, of the manuscript titled "Multi-spiral laser patterning of azopolymer thin films for generation of orbital angular momentum light" represented interesting results in the field of patterning the optically semi-transparent azopolymer films at the wavelength of 532 nm by using two-beams (an OV and a Gaussian beam with a spherical wavefront) interference lithography for realization spiral-mass transfer with the desired number of the formed spirals via applying laser radiation from a solid-state laser source (λ = 532 nm, Pout = 20 mW).

I would like to thank the authors they took into account the recommendations done.

About the remark 6 : “Did the authors notice some influence of the glass substrate on the properties of the patterned azopolymer films during the laser patterning by the OV beam and later, for instance some modification and/or activation of the opposite surface of the film, namely of the film interface-substrate or on the glass surface? This could happen when an optically transparent film deposited on glass or other type of substrate is irradiated by laser beam. Even more sometimes this is the desired effect, but sometimes it is strongly unwanted effect.”

 Just for clarity, I would like to specify that I mean reflection of the laser beam from the glass substrate surface to the film, which could cause some thermal effects, and/or this reflected beam to interfere with the incident beam (which is transmitted, since the optical transmittance of the film is about 50% at 532 nm) and thus to cause modifications on the opposite side of the layer. So, the authors could take into account this clarification when explain the influence of the glass substrate in the text between the Lines 206-217. Also, they could include in the manuscript the optical transmittance spectrum of the film for confirming their statement in Lines 206-217.

Author Response

We are thankful to reviewers for their useful comments and suggestions, which allow us to improve the quality of the manuscript making it more clear for readers. We believe the corrections made address the Reviewer’s concerns making the manuscript suitable for publication in the journal.

All changes in the manuscript are highlighted by green color.

Reviewer 1

The authors Alexey P. Porfirev, Nikolay A. Ivliev, Sergey A. Fomchenkov, Svetlana N. Khonina, of the manuscript titled "Multi-spiral laser patterning of azopolymer thin films for generation of orbital angular momentum light" represented interesting results in the field of patterning the optically semi-transparent azopolymer films at the wavelength of 532 nm by using two-beams (an OV and a Gaussian beam with a spherical wavefront) interference lithography for realization spiral-mass transfer with the desired number of the formed spirals via applying laser radiation from a solid-state laser source (? = 532 nm, Pout = 20 mW).

I would like to thank the authors they took into account the recommendations done.

About the remark 6 : “Did the authors notice some influence of the glass substrate on the properties of the patterned azopolymer films during the laser patterning by the OV beam and later, for instance some modification and/or activation of the opposite surface of the film, namely of the film interface-substrate or on the glass surface? This could happen when an optically transparent film deposited on glass or other type of substrate is irradiated by laser beam. Even more sometimes this is the desired effect, but sometimes it is strongly unwanted effect.”

 Just for clarity, I would like to specify that I mean reflection of the laser beam from the glass substrate surface to the film, which could cause some thermal effects, and/or this reflected beam to interfere with the incident beam (which is transmitted, since the optical transmittance of the film is about 50% at 532 nm) and thus to cause modifications on the opposite side of the layer. So, the authors could take into account this clarification when explain the influence of the glass substrate in the text between the Lines 206-217. Also, they could include in the manuscript the optical transmittance spectrum of the film for confirming their statement in Lines 206-217.

Reply:

Thank you for your comment. We added Figure 1 demonstrating the optical transmittance spectrum of the used azopolymer thin film. In addition, we added the following text:

In addition, we did not detect any modification of the film interface-substrate or on the glass surface. This was most likely caused by the relatively low power of the laser radiation used.